# Algorithms for Finding Vulnerabilities and Deploying Additional Sensors in a Region with Obstacles

Kibeom Kim ⬢ and Sunggu Lee *⬢

Department of Electrical Engineering, Pohang University of Science and Technology (POSTECH), Pohang 37673, Korea; kimpp200@gmail.com
* Correspondence: slee@postech.ac.kr; Tel.: +82-54-279-2936

**Abstract:** Consider a two-dimensional rectangular region guarded by a set of sensors, which may be smart networked surveillance cameras or simpler sensor devices. In order to evaluate the level of security provided by these sensors, it is useful to find and evaluate the path with the lowest level of exposure to the sensors. Then, if desired, additional sensors can be placed at strategic locations to increase the level of security provided. General forms of these two problems are presented in this paper. Next, the minimum exposure path is found by first using the sensing limits of the sensors to compute an approximate "feasible area" of interest, and then using a grid within this feasible area to search for the minimum exposure path in a systematic manner. Two algorithms are presented for the minimum exposure path problem, and an additional subsequently executed algorithm is proposed for sensor deployment. The proposed algorithms are shown to require significantly lower computational complexity than previous methods, with the fastest proposed algorithm requiring $O(n^{2.5})$ time, as compared to $O(mn^3)$ for a traditional grid-based search method, where $n$ is the number of sensors, $m$ is the number of obstacles, and certain assumptions are made on the parameter values.

**Keywords:** surveillance; minimum exposure path; sensor deployment; target discovery; wireless sensor network (WSN)



## 1. Introduction

A wireless sensor network (WSN) can be used for various tasks such as monitoring the environment and industrial facilities. Each generation of sensors tends to be smaller, more energy efficient, and less expensive than the previous generation. These trends have permitted the development of systems with an extremely large number of sensors. Management of such large networks entails solving problems related to how sensors are deployed, interconnected, and communicate with each other [1].

An important application of sensor networks and WSNs is surveillance [1], which can be conducted using smart networked surveillance cameras (a type of expensive networked sensor device) or simple sensor devices (such as heat, sound, or range sensors). For this type of application, the degree of security provided by the WSN is determined mainly by the positioning of the sensors. A common method for evaluating the adequacy of this type of WSN is to compute the minimum exposure path, which is a path that traverses the guarded region with the lowest level of exposure (Section 3.2) to the sensors. Then, if the resulting level of security is deemed to be insufficient, the existing sensors can be repositioned, additional sensors can be deployed, or both.

This paper targets the problem of WSN surveillance in a rectangular region with obstacles and sensors that have sensitivities that vary depending on the distance to the target being detected [2–8]. The level of security provided by the sensors can be evaluated by finding the minimum exposure path in this target region. This paper proposes a general model and solution for this problem. In the proposed method, the outer limits of the sensing ranges of the sensors and a segment tree data structure are used to compute a feasible area, within which the minimum exposure path must reside. Then, a grid is used

within this feasible area to search for the minimum exposure path. Given $n$ sensors, $m$ obstacles, and a few practical assumptions on model parameter values, the computational complexity of the proposed algorithm is $O(n^{2.5})$, which is a significant improvement over previous approaches, with computational complexity $O(mn^3)$.

In addition to computing the minimum exposure path, the proposed algorithm can be used to determine the location (and pointing direction, if appropriate) where an additional sensor can be placed to reduce the vulnerability of the target region. This can easily be done since the process for computing the minimum exposure path includes the determination of the feasible area for this minimum exposure path, and the feasible area is the most vulnerable area [9–11]. The main contributions of this paper are as follows.

- A general model, which encompasses most of the previously proposed models, is proposed for the minimum exposure problem.
- A new algorithm is proposed for the minimum exposure problem. For large $n$ and $m$ values, the proposed algorithm is a significant improvement over previous approaches.
- A new near-optimal algorithm is proposed for proper sensor deployment.

The rest of this paper is organized as follows. Section 2 reviews previous related research work. In Section 3, the problem being addressed in this paper is formally modeled and the proposed solution approach is described. In Section 4, the proposed solution and previous possible solutions are compared analytically, and an efficient sensor deployment algorithm is presented. Finally, concluding remarks are provided in Section 5.

## 2. Related Work

### 2.1. Voronoi Diagram and Delaunay Triangulation

The Voronoi diagram [12] is a graph consisting of edges created by drawing dividing lines between each pair of adjacent sensors such that each line is equidistant to the two sensors that it separates, with the line being terminated when it intersects with another line or the outer boundary of the region being monitored. The Delaunay triangulation is the dual graph of a Voronoi diagram. These two graph theory concepts have been found to be useful for computing minimum and maximum exposure paths given certain assumptions on the sensitivities of the sensor used.

Marengoni et al. [1] describe research work on surveillance in a given area referred to as an art gallery. Given that the entire art gallery should be covered by at least one sensor, the authors address the problem of the minimum number of sensors that need to be deployed. The sensors are modeled as disks with infinite ranges. Even though it only considers a two-dimensional (2-D) area, if there are Steiner points, the coverage problem becomes NP-hard [1]. Otherwise, it is NP-complete. To solve the same problem in a 3-D space, the authors of [1] suggest the use of a polynomial approximation solution, which adopts a greedy approach using Delaunay triangulation and graph coloring concepts [12].

Meguerdichian et al. [13,14] suggest an approach that applies a geometrical algorithm to the coverage problem. They assume all $n$ sensors are modeled with identical disks in which the sensitivity decreases as the distance between a point and a sensor increases. They define a maximal breach path as a path in which the distance from the closest sensor and itself is kept at a maximum. They also define a maximal support path as a path in which the distance from the closest sensor and itself is kept at a minimum value. To find the maximal breach path, they construct the Voronoi diagram [12]. After that, they set the weight of an edge in the Voronoi diagram to be the distance between that edge and the nearest sensor. Finally, they find the maximal breach path using a binary search process. The key value used is the arithmetic average of the biggest and smallest values of the edges. The edges with weights over the key value are inserted in $G'$, and Breadth-First-Search (BFS) is conducted on $G'$. If BFS is successful, the key value is increased following the binary search. Otherwise, the key value is decreased following the binary search. After renewal of the key value, $G'$ is reinitialized and edges that are bigger than the key value are inserted in $G'$. This process is continued until the difference between two consecutive key values is less than a tolerance value. The maximal support path is found in a similar manner

but with a Delaunay triangulation graph. The maximal breach path can be found by first constructing the Voronoi diagram in $O(n \log n)$ time. The number of nodes and edges of the Voronoi diagram is $O(n)$, so each BFS takes $O(n)$ time and the binary search performs BFS in $\log(n)$ time. Therefore, the total time complexity becomes $O(n \log n)$.

Later, using similar definitions and assumptions, Li et al. [15] suggested computing the shortest path using the sums of the weights of the edges, not the minimum weights of edges as in a maximal breach path. To find the shortest path, they used the Bellman–Ford algorithm. However, if the edge values are nonnegative, Dijkstra's algorithm [16], with computational complexity $O(n \log n)$, is faster than the Bellman–Ford algorithm, with computational complexity $O(n^2)$.

Hou et al. [17] suggested another method that produces both the maximal breach and maximal support paths using the Gabriel graph [12] and Dijkstra's algorithm. A mathematical proof was introduced to show the correctness of their result.

Mehta et al. [18] defined a breach number tree and introduced Kruskal's algorithm [12] to compute the maximal breach path and maximal support path. Using their method, if the Voronoi diagram with weight edges is constructed first, the paths can be computed in $O(|P|)$, where $|P|$ is the number of edges of the paths, and the values of the paths can be computed in $O(1)$.

In general, given $n$ sensors, the computation of the Voronoi diagram itself can be completed in $O(n \log n)$ time because the $x$ and $y$ coordinates of all sensors can be sorted first to determine the pairs of sensors to consider in the creation of the Voronoi diagram. The number of vertices $|V|$ and edges $|E|$ in the Voronoi diagram are $O(n)$, so a shortest-path search algorithm, such as Dijkstra's algorithm, could be used to compute the maximal breach path in time $O(|V|^2)$ or $O(|E| + |V| \log |V|)$ time using a min-priority queue implemented by a Fibonacci heap [16]. Thus, the overall time complexity for the computation of the maximal breach path using a Voronoi diagram is $O(n \log n)$.

### 2.2. Effect of Multiple Sensors Along a Path

Although the Voronoi diagram and Delaunay triangulation are useful concepts, they are found to be insufficient for the computation of the exact minimum exposure paths when the combined effect of multiple sensors are considered. For example, Figure 1 shows that if multiple sensors are clustered together in the lower region of an area while only one sensor is located in the upper region, then the minimum exposure path will be above the set of edges in the Voronoi diagram for this area. A detailed description of this effect is shown in Section 3.2 with an experiment based on the formal definitions of sensors, discover function, and exposure function.

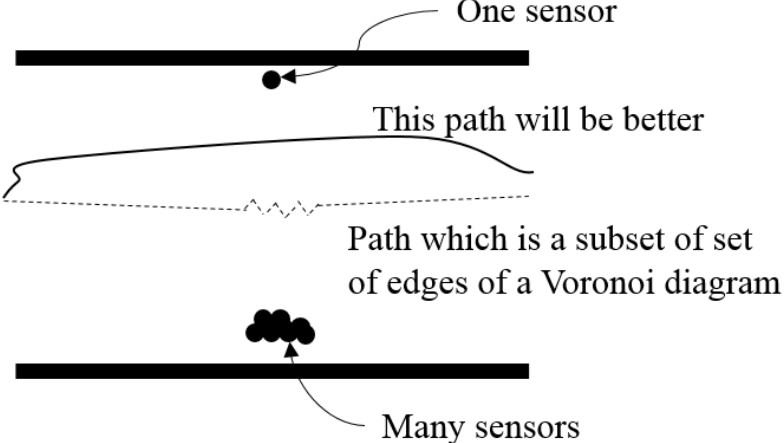

**Figure 1.** An example showing the shortcomings of a minimum exposure path found with a Voronoi diagram.

For more accurate path computation, the authors of [18–21] suggested a k-support path problem, and the authors of [19–21] solved this problem by implementing a k-th Voronoi diagram that includes the effect of the first to k-th nearest sensors. The authors of [19–21] defined the k-th nearest point Voronoi diagram (KNP-Voronoi diagram). Constructing the KNP-Voronoi diagram takes $O(k^2 n \log n)$ time and computing the minimum k-support path using Dijkstra's algorithm takes $O(k^2 n \log n)$ time. The authors of [22] found the maximal support path in the map, including the effect of not only the $n$ sensors but also $m$ obstacles. The authors of [22] suggested the use of the Constrained and Weighted Voronoi diagram (CW-Voronoi diagram). According to [23], the CW-Voronoi diagram has $O(m^2 n^2 + n^4)$ vertices and edges, and constructing the CW–Voronoi diagram takes $O(m^2 n^2 + n^4)$ time. Finally, running Dijkstra's algorithm on these edges requires $O((m^2 n^2 + n^4) \log(m^2 n^2 + n^4)) = O((m^2 n^2 + n^4) \log(mn + n^2))$ time.

A *discover* function can be defined as

$$discover : X \times Y \to \mathbb{R}$$

$$discover(x, y) = \max_{s \in S} f_s(x, y)$$

or as a circle with decreasing sensitivity as distance increases, so the discover value is usually defined as

$$discover(x, y) = \max_{s \in S} distance(s, (x, y)).$$

The maximum covered-distance of a path is defined with the definition of a path in Section 3.1,

$$cover1(p) = \max discover(p.x, p.y),$$

and the path $p$ that maximizes the discover value is called the best coverage path or the best support path. Similarly, the minimum covered-distance can be defined as

$$cover2(p) = \min discover(p.x, p.y)$$

and the path $p$ that minimizes the cover value is called the minimum coverage path or the maximal breach path.

The authors of [2,3] defined a sensor as a circle and found an optimal path based on intensity and exposure. The authors defined a discover function as the sum of all sensitivity potential function values at $(x,y)$ for all sensors,

$$discover : X \times Y \to \mathbb{R}$$

$$discover(x, y) = \sum_{s \in S} f_s(x, y),$$

exposure function,

$$exposure(p) = \int_p discover(s)ds = \int_{t_1}^{t_2} discover(p(t)) \left| \frac{dp(t)}{dt} \right| dt, \tag{1}$$

and an optimal path is the path that minimizes the exposure value. The authors first found an optimal path with one sensor. If the starting point is $(1, 0)$, the goal is $(0, 1)$, there is a sensor $s = (0, 0, \frac{1}{\sqrt{x^2+y^2}}, f_s)$, and $f_s$ is the same as Equation (5) with $\vec{d} = (1, 0)$ and $\theta = 2\pi$, then the optimal path is $(\cos \frac{\pi}{2} t, \sin \frac{\pi}{2} t)$ and the exposure along this path is $\frac{\pi}{2}$. Based on the optimal path for a single sensor, they suggested a heuristic method for finding an optimal path with several sensors by running the Dijkstra algorithm map divided into horizontal, vertical, and diagonal grid lines. The authors of [4] defined a sensor as a cone and proposed a heuristic optimal path search method with several sensors by running Dijkstra's algorithm in a map divided into horizontal and vertical grid lines.

Later, approaches were proposed for finding an optimal exposure path. In [24,25], for a single sensor map, an optimal exposure path was calculated by solving the Euler–Lagrangian equation in polar coordinates, while setting $\rho$ and $\phi$ as independent variables. The result is exactly the same as [2,3]. The authors of [26] used a similar method, but the optimal exposure path was calculated by solving the Euler–Lagrangian equation in Euclidian coordinates with $x$ and $y$ used as independent variables. For a map with multiple sensors, the authors of [24–26] used timing segmentation. They calculated the optimal direction for a specific position for an object, moved it slightly, calculated it again, moved it slightly, and so on until the object arrived at the goal. The authors of [27] set $y$ as a function of $x$ and solved the Euler–Lagrangian equation with $x$ used as an independent variable. For multiple sensors, the authors of [27] considered only neighboring sensors and found an approximated path, which was a localized minimum exposure path.

The sensitivity potential function can be modeled as varying according to the target velocity, $\frac{d}{dt}p(t) = (\frac{dx}{dt}, \frac{dy}{dt})$ [24–26]. In addition, the probability of detection will decrease if the target accelerates. Consider a sensor modeled as $s = (0, 0, \frac{1}{\sqrt{x^2+y^2}} \frac{1}{\sqrt{(\frac{dx}{dt})^2+(\frac{dy}{dt})^2}}, f_s)$, where $f_s$ is the sensitivity of the sensor, which varies depending on the sensing location and the velocity of the target. Then the *exposure* can be modeled as follows.

$$exposure = \int_{t_1}^{t_2} \frac{1}{\sqrt{x^2+y^2}} \frac{1}{\sqrt{(\frac{dx}{dt})^2+(\frac{dy}{dt})^2}} \sqrt{(\frac{dx}{dt})^2+(\frac{dy}{dt})^2} dt = \int_{t_1}^{t_2} \frac{1}{\sqrt{x^2+y^2}} dt$$

and the Euler–Lagrangian equation becomes

$$\frac{\partial}{\partial x} \frac{1}{\sqrt{x^2+y^2}} = 0, \text{ and } \frac{\partial}{\partial y} \frac{1}{\sqrt{x^2+y^2}} = 0,$$

and the solution is

$$x^2 + y^2 = \infty, \tag{2}$$

which is not feasible. Moreover, the Euler–Lagrangian equation does not provide a solution when the discover function does not have a continuous first partial derivative.

Creating a grid on the map and running the Dijkstra algorithm with that grid is an alternative practical solution for general maps. The authors of [5] divided a map with square grids and introduced Gaussian noise in grid edges. After the values of grid edges were set, they found an optimal path with the Dijkstra algorithm. The authors of [28] defined a mathematical model of physarum, used physical features of physarum and found a path using these features. The authors of [6–8] not only set the value of grid edges as probabilistic values but also defined an exposure value in a probabilistic manner as follows.

$$discover : X \times Y \to \mathbb{R}$$

$$discover(x, y) = 1 - \prod_{s \in S}(1 - f_s(x, y)),$$

Furthermore, the $\Re$ multiplications can be changed to $\Re$ additions by converting the values to their logarithms. Then, $\Re$ additions can be used to calculate the grid edge values in the integral.

$$exposure(p) = exp(\int_p \ln(discover(s))ds) \tag{3}$$

After that, they used the Dijkstra algorithm to find the shortest path.

The authors of [29] use both the Voronoi diagram and the exposure concept. They used the Voronoi diagram but then corrected the path by considering the exposure to neighboring sensors. The authors of [30] then proposed an improvement by defining

intensity as the sensing value of the nearest sensor and then integrating the effect of all sensors.

### 2.3. Sensor Deployment

The authors of [2,3,6,8,13,14,17,30–32] defined sensors that have circle-shaped sensitivity functions. The authors of [13,17] deployed sensors in the middle of the least weighted edge of the Voronoi diagram. The authors of [2,3] conducted an experiment that compared random deployment and deterministic deployment. In deterministic deployment, they deployed sensors in the middle sections of their grid edges, which can be formed into triangle or hexagon shapes by changing the definitions of sensors and the discover value. Reference [13] also divided the target map into grids and formulated the one-level coverage problem, and then proposed a heuristic algorithm called Tabu Search (TS). The authors of [30] deployed sensor networks based on Average Linear Uncovered Length (ALUL) and Minimum Linear Uncovered Length (MLUL) definitions. They divided the map of grid edges and selected the position of the additional sensor that set the ALUL or MLUL to be the maximum. For MLUL, they used the Voronoi diagram. The authors of [6,8] deployed some of the sensors randomly and checked whether the minimum exposure satisfied the desired level. If not, more sensors were added and the previous step was repeated. This was continued until the minimum exposure satisfied the desired level. Reference [14] deployed an additional sensor in the middle of the grid edge square in which the summation of the edges was the smallest. Reference [31] deployed sensors randomly the first time and then let each sensor have a virtual repulsive force to other sensors so that they pushed each other apart. The authors of [33] also deployed sensors with the virtual force for minimizing the required number of mobile sensors in sweep coverage under the energy constraint. The authors of [32] presented an algorithm for finding locally-optimal coverage in the minimum unit-disk cover problem through a Lagrangian minimum approach with $O(ln^4)$, where $l$ is the number of iterations and $n$ is the number of locations that must be included.

The authors of [9–11,34–37] defined sensors that have cone shaped sensitivity functions, which is a more general form than a circle. The authors of [34–37] deployed sensors in a region with obstacles. The authors defined a mathematical model for sensors in which the sensitivity extends to the boundary of the region or obstacles. The authors of [34,35] deployed sensors with a greedy algorithm. Reference [36] divided a region into grids and suggested a solution based on solving the integer nonlinear programming problem. As this becomes more difficult when the problem size gets larger, the authors of [37] suggested an immune-based algorithm. The authors of [38] suggested a barrier sweep coverage problem for covering not a whole area but finite critical curves with sector type sensors. They proved the barrier sweep coverage problem is an NP-hard problem and proposed an approximate algorithm.

The authors of [9,10] defined the most general form of the sensor, which has its own sensing radius and directional vector. They used virtual repulsive force and calculated the directional vector by virtual force. The authors of [11,39] applied Particle Swarm Optimization (PSO) to the coverage problem with this sensor model, and Reference [40] suggested Hybrid Genetic Particle Swarm Optimization (H-GPSO), which is a mixing model of the models in Reference[28] and [11]. The region considered in [9–11] does not include obstacles.

## 3. Preliminaries

### 3.1. System Model

A general model is presented for the computation of the minimum exposure path problem. This model will attempt to encompass almost all of the previous models used while addressing their shortcomings.

We assume that the region to be guarded is a 2-D rectangle of dimensions $B_X$ and $B_Y$. $n$ sensors $s_i \in S$ are positioned within this region to detect trespassers, but the approach

used here can easily be extended to a 3-D arbitrarily-shaped guarded region by replacing 2-D vectors to 3-D vectors. Each sensor $s_i \in S$ is positioned at $\vec{s}_{i,pos} = (x_i, y_i)$ and has a sensitivity potential function $f_{s_i}(\vec{x}_{pos})$ that denotes the strength of the signal detected at sensor $s_i$ due to a trespasser located at $\vec{x}_{pos} = (x, y)$. Depending on the function $f_{s_i}(\vec{x}_{pos})$, we can model a sensor for which an event within a radius of $(x_i, y_i)$ is always detected whereas all events farther than the radius from $s_i$ remain undetected (the commonly-used solid disk model). More realistically, an $f_{s_i}(\vec{x}_{pos})$ function that decreases with the distance to $\vec{x}_{pos} = (x, y)$ can be used to model a sensor in which the sensitivity to an event diminishes as the distance to the event is increased. Finally, this type of general definition for $f_{s_i}(\vec{x}_{pos})$ can also be used to model a sensor that can monitor events within a cone shape, such as would be the case with a camera with a limited field of view.

The presence of a set $W$ of impassable obstacles is assumed, such as pillars, facilities in a factory, or a great pile of luggage. For an obstacle $w \in W$, a function $f_w$ exists that describes the obstacle as $w = \{(x, y) \mid (x, y) \in X \times Y$ and $f_w(\vec{x}_{pos}) > 0\}$. The function *Boundary*(*area*) is used to denote the boundary of a given area. For example, given $f_w$ as below,

$$f_w(\vec{x}_{pos}) = \begin{cases} 1 \text{ if } 3 < x < 5 \text{ and } 1 < y < 2, \\ 0 \text{ otherwise} \end{cases} \tag{4}$$

obstacle $w$ is a rectangular area (Section 3.3). Thus, *Boundary*($w$) = $\{(3, y) \mid 1 < y < 2\} \cup \{(5, y) \mid 1 < y < 2\} \cup \{(x, 1) \mid 3 < x < 5\} \cup \{(x, 2) \mid 3 < x < 5\}$. If the sensing area occupies the back of the obstacle, it means that the sensor has a field of view on the back of the obstacle, so it implies the situation should be a 3-D model rather than a 2-D model. Therefore, it is assumed that there is no field of view behind the obstacle from the 2-D sensor. For example, Figure 2 shows a university dormitory area with four CCD (Charge Coupled Device) cameras and one dormitory building. A thief is at the start location, and the thief wants to get to the goal location while avoiding the sensing area of CCD cameras as much as possible. As the dormitory building is higher than any camera height, the camera's field of view cannot reach behind the building (Figure 2).

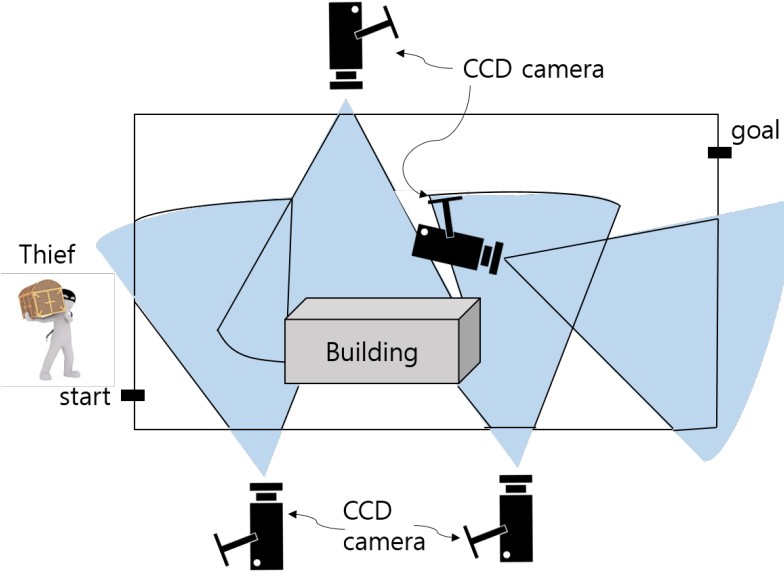

**Figure 2.** Sensing areas from an arrangement of CCD cameras (blue cones) and a building (gray cuboid).

The sensitivity potential function for a sensor $s \in S$ is assumed to have the following general form,

$$f_s(\vec{x}_{pos}) = \begin{cases} \dfrac{K_s}{|\vec{x}_{pos} - \vec{s}_{pos}|^{\frac{l_1}{2}} \left|\frac{d\vec{x}_{pos}}{dt}\right|^{\frac{l_2}{2}}} & \text{if } \arccos \dfrac{\vec{d}_s \cdot \vec{x}_{pos}}{|\vec{x}_{pos} - \vec{s}_{pos}|} < \dfrac{\theta_s}{2} \text{ and } \dfrac{K_s}{|\vec{x}_{pos} - \vec{s}_{pos}|^{\frac{l_1}{2}} \left|\frac{d\vec{x}_{pos}}{dt}\right|^{\frac{l_2}{2}}} > \eta, \\ 0 & \text{otherwise} \end{cases} \tag{5}$$

where $\eta \in \mathbb{R}$ is the noise potential, $\vec{d}_s$ is a directional vector, and $\frac{\theta_s}{2}$ is the angle of the field-of-view; $K_s$, $l_1$, and $l_2$ are sensor-dependent constants (Figure 3). The given coordinate $\vec{x}_{pos}$ and position of sensor $s_i$ $\vec{s}_{i,pos}$ are determined according to the dimension of the guaranteed region, which is $(x, y)$ and $(x_i, y_i)$ for 2-D (Figure 3a) and $(x, y, z)$ and $(x_i, y_i, z_i)$ for 3-D (Figure 3b).

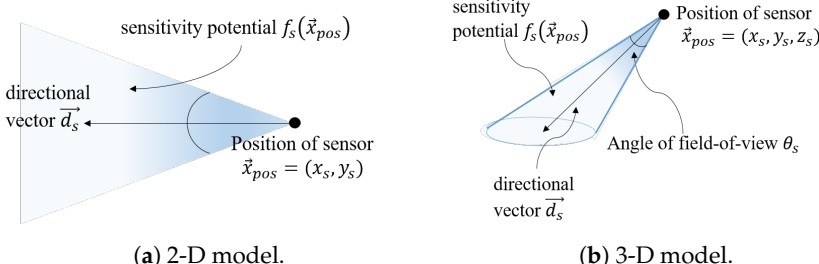

(**a**) 2-D model.　　　　　　(**b**) 3-D model.

**Figure 3.** Sensitivity potential function.

In addition, we adopt the definitions for  coverage and  sensing area used in [41] and generalized the definitions with obstacles.

**Definition 1.** *A position $\vec{x}_{pos}$ is* covered *by a sensor s if $f_s(\vec{x}_{pos}) > 0$ and $f_w(\vec{x}_{pos} = 0$. A sensing area S is a set of positions that are covered by exactly the same set of sensors.*

To prevent a target from penetrating into obstacles, we define the sensitivity potential in obstacles to be positive infinity. In $X \times Y$, we should construct the appropriate data structure using $W$ and $S$. The appropriate data structure is one for which, when a coordinate is given, the information about which sensor functions or obstacles are covered is returned. We assume that sensors cannot see the target when it is obscured by obstacles. As an example, this assumption and Definition 1 results in sensing areas $S1$ to $S11$ when the sensors and an obstacle are arranged as in Section 3.3, which is geometrically simplified from the realistic scenario in Figure 2.

Finally, the area can be defined formally. The position where the target starts to move, or the entrance of the map, is denoted as $(x_{in}, y_{in}) \in X \times Y$, and the position where the target stops moving, or the exit point of the map, is denoted as $(x_{out}, y_{out}) \in X \times Y$. The map $M$ is denoted formally as $(X, Y, S, W, (x_{in}, y_{in}), (x_{out}, y_{out}), \eta)$, where $\eta \in \mathbb{R}$ is the noise potential.

### 3.2. Discover and Exposure Functions

The target wants to move from $(x_{in}, y_{in})$ to $(x_{out}, y_{out})$ with the least chance of being discovered by a sensor. To formally find the movement path of the target, we can consider a real analytic function $p$ from the time interval to a specific point in the map, $p : \mathbb{R} \to X \times Y$. However, the paths we consider start at $(x_{in}, y_{in})$ and end at $(x_{out}, y_{out})$. To avoid the infinite velocity value that results from the Euler–Lagrangian Equation (2), we set $\left| \frac{dp(t)}{dt} \right| = 1$. Finally, the set of paths is formally denoted as $P = \{ p \mid$ , $p$ is real analytic, the initial point of $p$ is $p_{start}$, and the final point of $p$ is $p_{goal} \}$. A path $p$ is an element of $P$.

The calculations using Equations (1) and (3) to set the optimal path are similar. Eventually they calculate the discover value by summing several functions and then calculate exposure as the integral of the discover function; the minimum paths are the same because the exponential function is increasing. For this reason we select Equation (1) for our target calculations.

**Definition 2.** *A* discover *value about specific position $\vec{x}_{pos}$ is $discover(\vec{x}_{pos}) = \sum_{s \in S} f_s(\vec{x}_{pos})$. An* exposure *value about specific path p is $exposure(p) = \int_{t_1}^{t_2} discover(p(t))dt$.*

We also set the path that has the smallest exposure value of map $M$ as $p_{min}(M)$. The $exposure(p_{min}(M))$ is the degree of vulnerability of a region, which can be defined mathematically as shown below.

Although the Voronoi diagram and Delaunay triangulation are useful concepts, they are found to be insufficient for the computation of the exact minimal exposure paths when the combined effect of multiple sensors is considered. For example, if multiple sensors are clustered together in the corner of the map with weighted values (Figure 4), then the minimal exposure path from the Voronoi diagram is not actually the minimum. To verify that, we set an experiment. From the definition of the sensor (Equation (5)), we set $\theta_s = 2\pi$, $K_s$ = 10,000, $l = 2$, and set the noise $\eta = 1$. If the sensors are deployed following Figure 4, the resulted minimum exposure path from the Voronoi diagram usually represents more exposure value even than the 50 × 50 grids diagram (Figure 5).

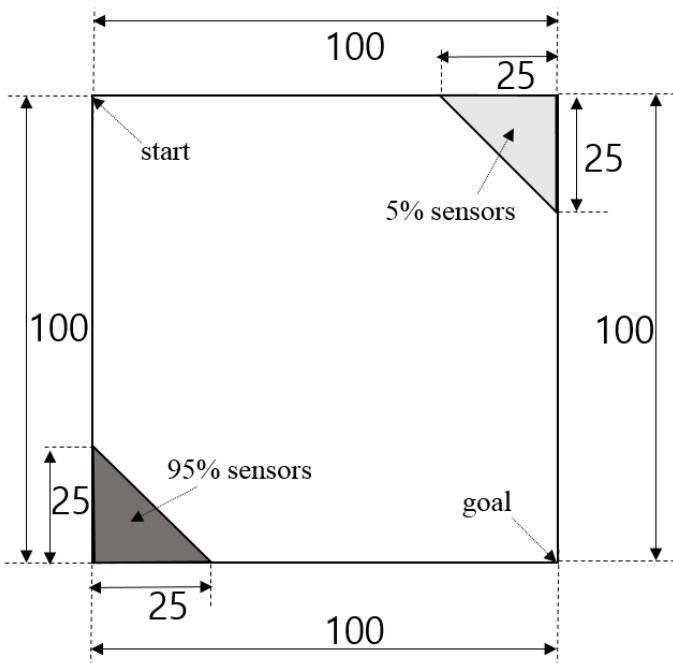

**Figure 4.** Weighted cluster deployment of sensors.

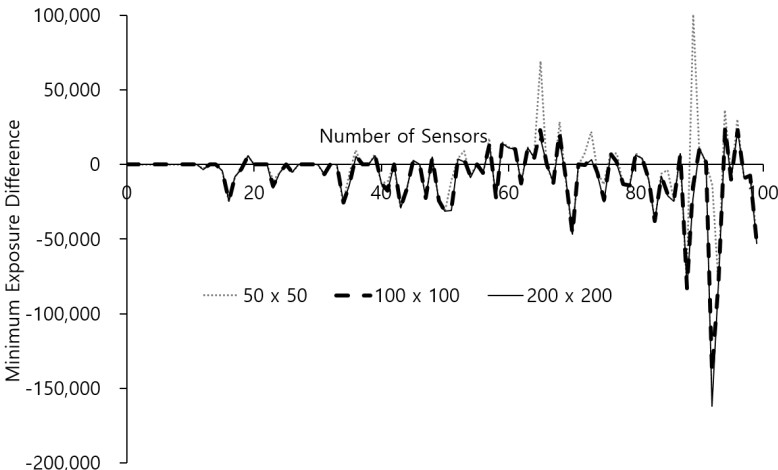

**Figure 5.** Minimum exposure difference from the minimum exposure value from the Voronoi Diagram (y) vs. number of deployed sensors following Figure 4. (x) in 100 × 100 area. Dotted line; 50 × 50 grids; broken line; 100 × 100 grids; solid line; 200 × 200 grids.

### 3.3. Feasible Paths and Areas

The proposed method reduces the time required to find the minimum exposure path. To accomplish this, we need to define additional sets and functions. Section 4 presents the actual algorithms and proofs that use these sets.

$\hat{p}_{min}(N, A)$ is the path found by running the Dijkstra algorithm following $N$ grid edges in area $A$. $\hat{p}_{min}(N, A)$ approaches $p_{min}(M)$ as $N$ goes to positive infinity,

$$p_{min}(M) = \lim_{N \to \infty} \hat{p}_{min}(N, M).$$

However, as $N$ increases, $\hat{p}_{min}(N, A)$ becomes more accurate but the number of computations increase. To reduce the number of computations with the same level of accuracy, we decided to limit the area of interest in the target map, and this area is called the *feasible area*, $\tilde{A}$. To find $\tilde{A}$, the skeleton of the feasible area, feasible paths, should be defined. A set of paths following the boundaries of sensing areas is denoted as $\bar{P} = \{p(t) \mid p \in P$ and $p \in Boundary(\text{whole sensing areas})\}$, and

$$\bar{p}_{min}(M) = arg \min_{p \in \bar{P}} exposure(p).$$

From $\bar{P}$, a set of feasible paths are denoted as $\tilde{P} = \{p(t) \mid p \in \bar{P}, exposure(\bar{p}_{min}) > \int_p (discover(s) - \min(R_{max}, radius(\text{sensing area that contains } s))|\nabla f(s)|)ds\}$ (Section 4.1), where $R_{max}$ is the maximum radius of the sensing areas. Finally, a feasible area $\tilde{A}$ is denoted as {region of expanding $\tilde{P}$ with the direction of $\nabla f(p(t))$ and $\min(R_{max}, radius(\text{sensing area which contains } p(t)))$ for all $p(t) \in \tilde{P}$}. For example, in Figure 6, if $P = \{p1, p2, p3\}$, then $\tilde{P} = \{p1, p2\}$. In Figure 7, if $\tilde{P} = \{p4, p5\}$, then $\tilde{A}$ would be the dark area shown in the figure.

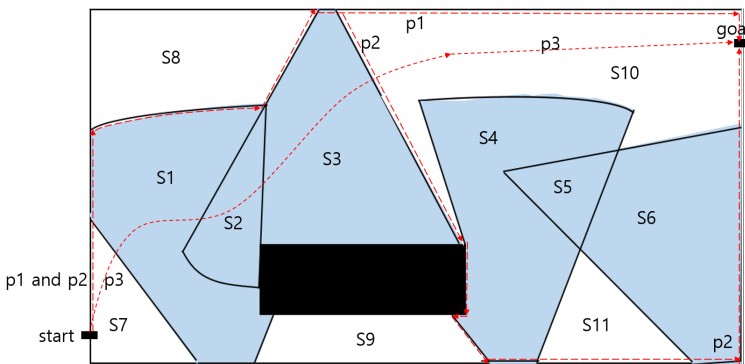

**Figure 6.** An example of $P$ (red dotted lines) and $\tilde{P}$.

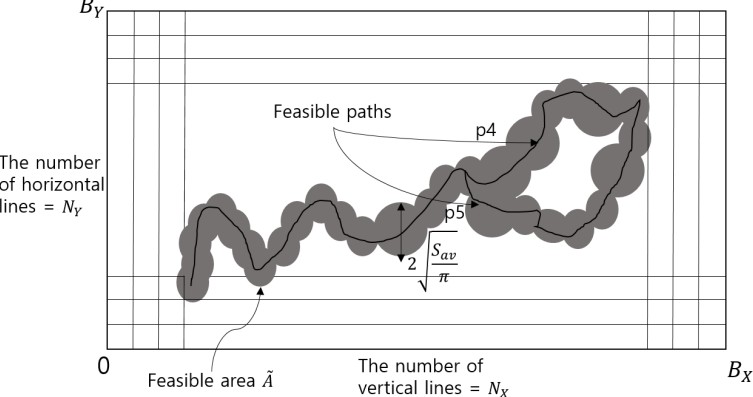

**Figure 7.** An example of $\tilde{P}$ and $\tilde{A}$.

To find $\hat{p}_{min}(N, M)$, grid edges should be created and processed. However, finding grid edges in only $\tilde{A}$, instead of the entire area $M$, obviously reduces the number of computations (Theorem 4). In addition, if creating $\tilde{A}$ does not increase the time complexity and $\hat{p}_{min}(N, M) = \hat{p}_{min}(N, \tilde{A})$, introducing the process for making $\tilde{A}$ is definitely acceptable (Theorem 3).

## 4. Solution and Analysis

### 4.1. Minimum Exposure Path

Definitions for *discover* and *exposure* functions were presented in Section 2.2. References [2–8] tried to find $p_{min}(M)$ by partitioning the map area using edges and weighting each edge using the exposure value along the edge. Finally, they ran the Dijkstra algorithm while following the grid edges (Algorithm 1).

This paper proposes a new, more efficient method for finding the minimum exposure path, as shown in psuedocode format in Section 4.1. In the proposed method, we want to reduce the time consumption by reducing the time required to calculate each grid edge, and reducing the number of grid edges to be considered. To reduce the time required to calculate each grid edge, we added the step "Construct sensing area data structure" and set the grid edge values "based on an efficient data structure" (Algorithm 2). For additional time reduction, we added the step "Make $\tilde{A}$" and "Run the Dijkstra algorithm with grid edges in $\tilde{A}$" (Algorithm 3).

---

**Algorithm 1:** Algorithm proposed by Liu in 2009 [4].

**Input:** map $M$
**Output:** approximate optimal path $\hat{p}_{min}(N, M)$
　1: Divide the entire map area with grid edges and set the values to consider all sensors and obstacles
　2: Run the Dijkstra algorithm with grid edges
　3: Construct sensing area data structure
　4: **return** $\hat{p}_{min}(N, M)$

---

**Algorithm 2:** An enhanced algorithm with segment trees.

**Input:** map $M$
**Output:** approximate optimal path $\hat{p}_{min}(N, M)$
　1: Construct sensing area data structure
　2: Divide the entire map area with grid edges and set the values from the sensing area data structure
　3: Run the Dijkstra algorithm with grid edges
　4: Construct sensing area data structure
　5: **return** $\hat{p}_{min}(N, M)$

---

**Algorithm 3:** The proposed minimum exposure path algorithm.

**Input:** map $M$
**Output:** approximate optimal path $\hat{p}_{min}(N, M)$
　1: Construct sensing area data structure
　2: Calculate edge values of *Boundary*(*sensing_areas*)
　3: Run the Dijkstra algorithm with *Boundary*(*sensing_areas*) and configure $\tilde{A}$
　4: Divide $\tilde{A}$ with grid edges and set the values from sensing area data structure
　5: Run the Dijkstra algorithm with grid edges in $\tilde{A}$
　6: **return** $\hat{p}_{min}(N, \tilde{A})$

---

To "Run the Dijkstra algorithm with $\bar{P}$" (Figure 8), the vertices and edges can be like those shown in Figure 8. The Dijkstra Algorithm can be used to find the shortest path using

the vertices and edges shown. We will show that Algorithms 1 and 3 return the same result, but that Algorithm 3 is faster than Algorithm 1.

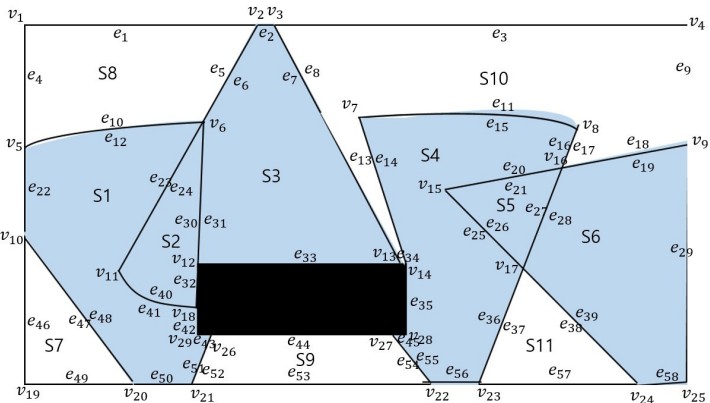

**Figure 8.** An example of constructing sensing area data structure.

**Theorem 1.** *Constructing the sensing area data structure takes $O((m + n)^2 \log(m + n))$ time*

**Proof of Theorem 1.** The sensing area data structure uses two concepts: a doubly-connected edge list and a segment tree. The doubly-connected edge list can be used to determine which sensors cover a specific region in $O(d_1)$ time if a boundary edge is given, where $d_1$ is the maximum number of sensors in a sensing area. The segment tree can be used to find a boundary edge in $O(\log(m + n))$ time if a point is given. When taken together, the list of sensors that cover a region can be found in $O(\log(m + n) + d_1)$ if a point is given. This is much faster than $O(m + n)$.

When a sensor is included in the map, intersections and the leftmost point and the rightmost point become vertices. A single line segment bifurcates into two edges if they have different exposure values. This rule has two exceptions: when the line segment is outside the map boundary and when it is inside an obstacle, because a target cannot go outside the map boundary or inside an obstacle (Figure 9). The edges created in this manner are stored in a segment tree (Figure 10). For each sensing area, information about edges, vertices, and the set of sensors that affect the sensitivity in that sensing area are stored.

The number of vertices, edges, and areas is $O((m + n)^2)$. Therefore, the sensing data structure can be constructed in $O((m + n)^2 \log(m + n)^2) = O((m + n)^2 \log(m + n))$ time. □

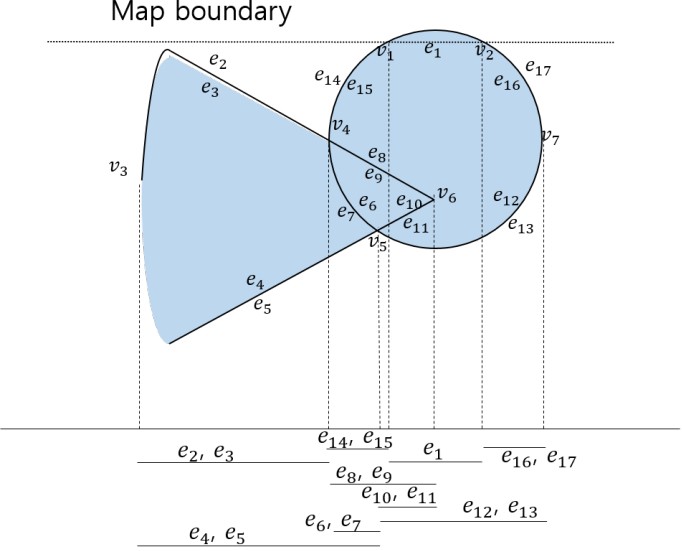

**Figure 9.** An example of segments to be stored in a segment tree.

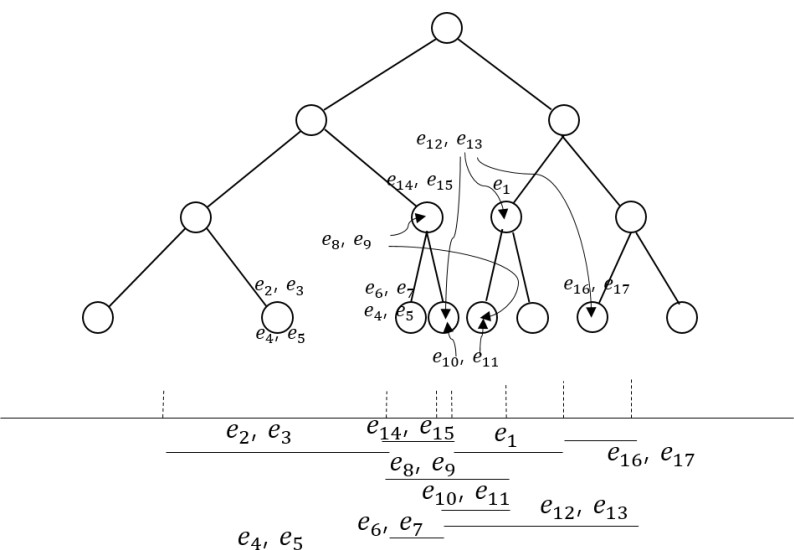

**Figure 10.** A segment tree storing the boundaries of sensing areas.

**Theorem 2.** *Algorithm 1 takes $O(N_X^2(mn + \log N_X))$ time and Algorithm 2 takes $O((m + n)^2 \log(m + n) + N_X^2(m + n + \log N_X))$ time.*

**Proof of Theorem 2.** If a query regarding point $(x, y)$ is given, the edges that reach $x$ are reported in $O(\log(m + n)^2 + d_2) = O(\log(m + n) + d_2)$ time, where $d_2$ is the number of reported edges. $d_2$ increases by two only when a sensing area covers the query point. Therefore, $1 \leq d_2 \leq (m + n)$. Each edge contains information about adjacent vertices and sensing regions. From the list of edges, the sensing region that contains the query point can be found in $O(d_2)$ time. Then the sensitivity is calculated in $O(d_1 + \log(m + n) + d_2)$ time. In a $k$-coverage map, $0 \leq k \leq d_1 \leq n$.

For each grid edge, the number of sensors to consider is $O(d_2)$ and finding a sensing area that contains the grid edge takes $O(\log(m + n)^2) = O(\log(m + n))$ time, so calculating the value of grid edge takes $O(d_1 + \log(m + n) + d_2)$ time.

In Algorithm 3, $\tilde{A}$ is the gray region in Figure 7. Grids are calculated only in the gray region in Figure 7. Let $N_X$ be the number of lines vertically, and $N_Y$ be the number of lines horizontally. The grid lines make a grid square. Then

$$\frac{B_Y}{N_Y} = \frac{B_X}{N_X} = \text{size of a side of a cell in the grid}$$

$$N_Y = \frac{B_Y}{B_X} N_X.$$

The number of vertices is $|V| = N_X N_Y = \frac{B_Y}{B_X} N_X^2 = O(N_X^2)$. The number of edges is $|E| = N_X(N_Y - 1) + (N_X - 1)N_Y = O(N_X^2)$, so $|V| = |E| = O(N_X^2)$ and the time complexity to calculate all edge values is $O(N_X^2(d_2 + \log(m + n) + d_1))$ and the time complexity to find the optimal path is $O(N_X^2 \log N_X^2) = O(N_X^2 \log N_X)$.

Therefore, the total time complexity is

$$O((m + n)^2 \log(m + n) + N_X^2(d_2 + \log(m + n) + d_1) + N_X^2 \log N_X)$$

$$= O((m + n)^2 \log(m + n) + N_X^2(d_2 + \log(m + n) + \log N_X + d_1).$$

Furthermore, by applying $0 \leq k \leq d_1 \leq n$ and $1 \leq d_2 \leq (m + n)$, the total time complexity of Algorithm 2 becomes $O((m + n)^2 \log(m + n) + N_X^2(m + n + \log N_X))$

Now we show why the "Construct sensing area data structure" step should be included. Previous work [2,3,24–27] did not include this step. If this step is ignored, all sensors and obstacles must be considered for all grid edges. The time complexity to cal-

culate values for all edge values is $O(N_X^2 mn)$, and to find $\hat{p}_{min}(N, M)$ is $O(N_X^2 \log N_X^2) = O(N_X^2 \log N_X)$. Therefore, the total time complexity of Algorithm 1 will be $O(N_X^2 mn + N_X^2 \log N_X) = O(N_X^2(mn + \log N_X))$. This is obviously bigger than $O(N_X^2(m + n + \log N_X))$. To find a more accurate path than the path found with a Voronoi diagram, $N_X \geq O(m + n)$. This makes $O(N_X^2(mn + \log N_X)) > O((m + n)^2 \log(m + n))$. Therefore, including the "Construct sensing area data structure" step reduces the necessary time complexity. $\square$

**Theorem 3.** *Algorithms 2 and 3 return the same path.*

$$\hat{p}_{min}(N, M) = \hat{p}_{min}(N, \tilde{A}).$$

**Proof of Theorem 3.** At first, the path can be filtered if the sensitivity function of each sensor is convex downward. To simplify the proof, we consider a fractional sensitivity potential model modeled by Equation (5) (Figure 11). For an arbitrary convex downward function $y = g(x)$, $g''(x) \geq 0$, and $g'(x + L) \geq g'(x)$ for all $x$ and $L > 0$. We set $h(x, L) = g(x + L) - (g(x) + Lg'(x))$, from which it follows that $\frac{h(x,L)}{\partial x} = g'(x + L) - g'(x) + Lg''(x) \geq 0$ and $h(x, 0) = 0$. Therefore, $h(x, L) \geq 0$ for all $x$ and $L > 0$.

Next, in a map, if we apply $y = ax + b$ in $\frac{1}{(x^2 + y^2)^k}$, it becomes $\frac{1}{\left((a^2 + 1)(x + \frac{b}{a^2 + 1})^2 + \frac{a}{a^2 + 1}\right)^k}$.

For this reason, any sensitivity potential projected in any direction is convex downward. No matter how many convex downward functions are added, the added function is also convex downward. For these two reasons, outside of $\tilde{A}$, no path has an exposure value lower than $\hat{p}_{min}(N, \tilde{A})$. $\tilde{A}$ always contains $\hat{p}_{min}(N, M)$. Moreover, $\tilde{A}$ contains $p_{min}(M)$ because the maximum distance between $\hat{p}_{min}(N, M)$ and $p_{min}(M)$ is less than the size of a side of a cell in the grid. $\square$

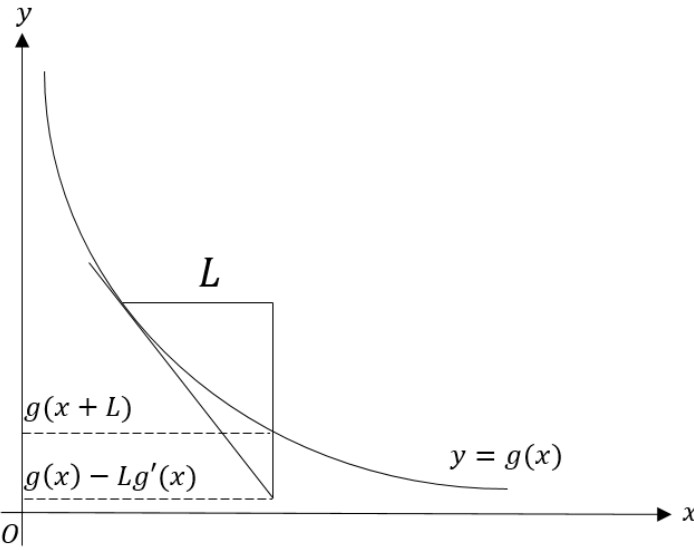

**Figure 11.** An illustration of the characteristics of a convex downward function.

**Theorem 4.** *The time complexity of Algorithm 3 is less or equal to that of Algorithm 2.*

**Proof of Theorem 4.** The parts in Algorithm 3 that perform more computations than Algorithm 2 are

1. Calculate edge values of *Boundary(sensing_areas)*;
2. Run the Dijkstra algorithm with *Boundary(sensing_areas)* and configure $\tilde{A}$.

The number of areas is $O((m + n)^2)$. Accordingly, the number of vertices and edges for "Run the Dijkstra algorithm with *Boundary(sensing_areas)* and configure $\tilde{A}$" are $|V| = O((m + n)^2)$, $|E| = O((m + n)^2)$. Then the Dijkstra algorithm takes $O((m + n)^2 \log(m +$

$n)^2) = O((m + n)^2 \log(m + n))$ time. Each of the result paths is stored including the sensor information for the path. In step "Divide $\tilde{A}$ with grid edges and set the values from sensing area data structure", grids can be configured through the sensor information. Therefore, those additional parts take $O((m + n)^2 \log(m + n))$ time. This is the same as the step "Construct the sensing area data structure", so this step does not increase the time complexity. For those reasons, the additional computations used in Algorithm 3 do not increase the time complexity. □

Although the above analysis has shown the time complexities of Algorithm 1 through Algorithm 3 in as exact a manner as possible, these time complexity equations can be expressed in simpler forms by adopting several practical assumptions.

First, for the reasonable coverage of the map, the number of sensors $n$ should be more than the number of obstacles $m$, and sensors almost cover the whole map. Then, the time complexity of Algorithm 2 becomes $O(n^2 \log n + N_X^2(n + \log N_X))$. Next, the shorter given path is, the smaller *exposure* of the path is. If a feasible path is about a straight line from the start in the lower left corner to the goal in the upper right, the area ratio of $\tilde{A}$ to whole map is about $\frac{2\sqrt{\frac{S_{av}}{\pi}}\sqrt{N_X^2 B_X^2 + N_Y^2 B_Y^2}}{N_X B_X N_Y B_Y}$, where $S_{av}$ is the average sensing area of sensors (Figure 7). According to the first assumption, $nS_{av} = N_X B_X N_Y B_Y$. This induces the number of vertices and edges in "Divide $\tilde{A}$ with grid edges and set the values from sensing area data structure" as

$$O(|V|) = O(|E|) = O\left(N_X N_Y \frac{2\sqrt{\frac{S_{av}}{\pi}}\sqrt{N_X^2 B_X^2 + N_Y^2 B_Y^2}}{N_X B_X N_Y B_Y}\right)$$

$$= O\left(\frac{\sqrt{\frac{N_X B_X N_Y B_Y}{n}}\sqrt{N_X^2 B_X^2 + N_Y^2 B_Y^2}}{B_X B_Y}\right) = O\left(\frac{N_X^2}{\sqrt{n}}\right).$$

Then, the time complexity to calculate all edge values is $O(\frac{N_X^2}{\sqrt{n}}(d_2 + \log(m + n) + d_1)) = O(N_X^2 \sqrt{n})$ and the time complexity to find the optimal path is $O(\frac{N_X^2}{\sqrt{n}} \log \frac{N_X^2}{\sqrt{n}}) = O(\frac{N_X^2}{\sqrt{n}} \log N_X)$. Therefore, the time complexity to find the optimal path is $O(n^2 \log n + N_X^2 \sqrt{n} + \frac{N_X^2}{\sqrt{n}} \log N_X)$. Finally, by considering the computation limits, the number of vertical grid lines $N_X$ can be set to $O(n)$, which derives the time complexity of Algorithms 1–3 as, respectively, $O(mn^3)$, $O(n^3)$, and $O(n^{2.5})$.

*4.2. Deployment of Additional Sensors*

Motivated by $\tilde{A}$, to handle the vulnerable area, additional sensors should be deployed in a manner that increases $exposure(p_{min}(M))$ as much as possible. The candidate points are near the locally maximum curvature points due to the definition of the sensitivity potential function. The points at which the curvature are locally maximum are stored in the set $\tilde{L}$. An example of this set of points is shown as $\tilde{L} = \{(\tilde{x}_1, \tilde{y}_1), (\tilde{x}_2, \tilde{y}_2), (\tilde{x}_3, \tilde{y}_3), (\tilde{x}_4, \tilde{y}_4), (\tilde{x}_5, \tilde{y}_5)\}$ in Figure 12.

For a point at which the curvature is a local maximum, such as $(\tilde{x}_j, \tilde{y}_j)$, where $0 \leq j \leq |\tilde{L}|$, the normal vector $\overrightarrow{N_j}$ can be computed. $\overrightarrow{d_j}$ is set to the opposite direction of $\overrightarrow{N_j}$, so $\overrightarrow{d_j} = -\overrightarrow{N_j}$ because the directional vector and normal vector are unit vectors. Let $(\bar{x}_j, \bar{y}_j)$ be the position of $s$ for the outer curve of $Boundary(\tilde{A})$, with the sensing area of $s$ being tangent to each other (Figure 13).

To compare the points in $\tilde{L}$, the maximum increment coverage value should be selected. However, the increment value is only considered in the line between $(\bar{x}_j, \bar{y}_j)$ and $(\tilde{x}_j, \tilde{y}_j)$ because the error is not considerable. Let $L_j$ be the distance between $(\bar{x}_j, \bar{y}_j)$ and $(\tilde{x}_j, \tilde{y}_j)$ (Figure 14). By following variable $\xi_j$ between $[0, L_j]$, the position $(x_j, y_j) = (\bar{x}_j, \bar{y}_j) + \frac{\xi_j}{L_j}\overrightarrow{d_j}$ varies. The outer interval of $[0, L_j]$ is not considered because the area of

($\tilde{A} \cap$ sensing area of $s$) by setting $(x_s, y_s) = (x_j, y_j)$ becomes smaller in the outer of the interval of the position of $s$. Therefore, the approximate maximum increment coverage value for $(\tilde{x}_j, \tilde{y}_j)$ is denoted as $incr_j$, and the corresponding $\xi_j$ is denoted as $\xi_{j,max}$.

$$incr_j = \max_{0 \le \xi_j \le L_j} \iint_{\substack{\tilde{A} \cap \text{sensing area of } s, \\ \text{where } (x_s, y_s) = (x_j, y_j)}} f_s(x, y) dx dy,$$

$$\xi_{j,max} = arg \max_{0 \le \xi_j \le L_j} \iint_{\substack{\tilde{A} \cap \text{sensing area of } s, \\ \text{where } (x_s, y_s) = (x_j, y_j)}} f_s(x, y) dx dy.$$

For each $(\tilde{x}_j, \tilde{y}_j)$, $incr_j$ is calculated and all locally maximum curvature points are compared based on the $incr_j$ value. The index for which $incr_j$ is the maximum value is denoted as $j_{max}$.

$$j_{max} = arg \max_{0 \le j \le |\tilde{L}|} incr_j.$$

From $j_{max}$, the undefined components of the additional sensor $s$ are now defined, $(x_s, y_s) = (x_{j_{max}}, y_{j_{max}})$, and $\vec{d_s} = \vec{d}_{j_{max}}$. After the sensor $s$ is included in $M$, $exposure(p_{min}(M))$ will be increased deterministically because the approximate position for the maximum increase is considered (Algorithm 4).

---

**Algorithm 4:** The proposed algorithm for deployment of an additional sensor.

---

**Input:** sensor $s$, map $M$
**Output:** new map $M$ including sensor $s$
1: Calculate $\tilde{A}$ and $Boundary(\tilde{A})$
2: Calculate $\tilde{L}$ and $|\tilde{L}|$
3: **for** each $(\tilde{x}_j, \tilde{y}_j) \in \tilde{L}$ **do**
4:　　Calculate $\vec{N_j}$ at $(\tilde{x}_j, \tilde{y}_j)$
5:　　$\vec{d_j} = -\vec{N_j}$
6:　　Calculate $(\bar{x}_j, \bar{y}_j)$, and $L_j$
7:　　Calculate $incr_j$ and $\xi_{j,max}$
8: **end for**
9: Calculate $j_{max}$, and fill the undefined components of $s$
10: Update sensing area with $s$
11: **return** $M$

---

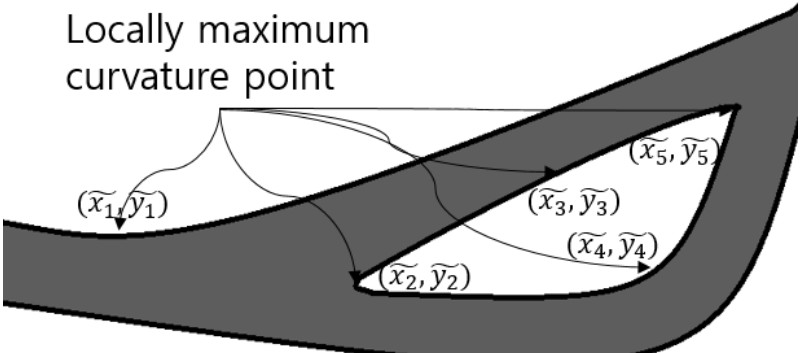

**Figure 12.** $(\tilde{x}_j, \tilde{y}_j)$ is one of the point at which the curvature is a local maximum in given $\tilde{A}$.

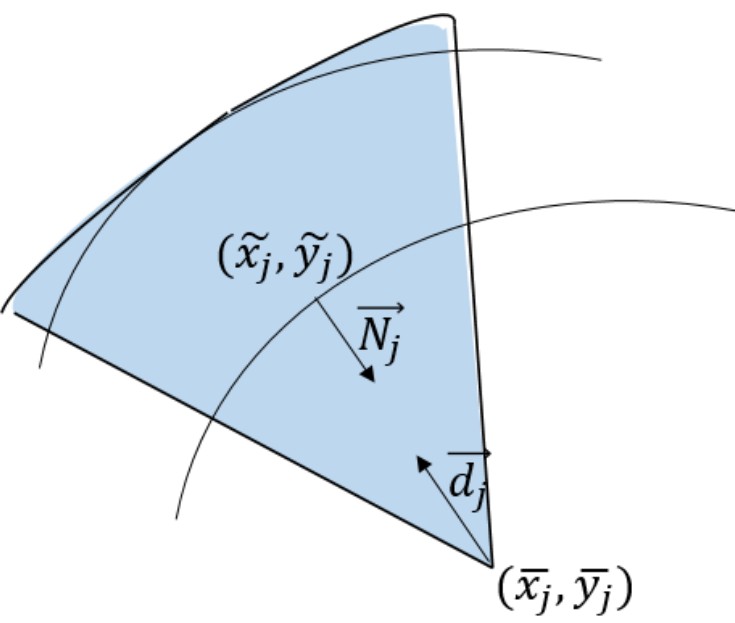

**Figure 13.** $(\bar{x}_j, \bar{y}_j)$ is the position of $s$ for outer curve of $Boundary(\tilde{A})$ and sensing area of $s$ being tangent to each other.

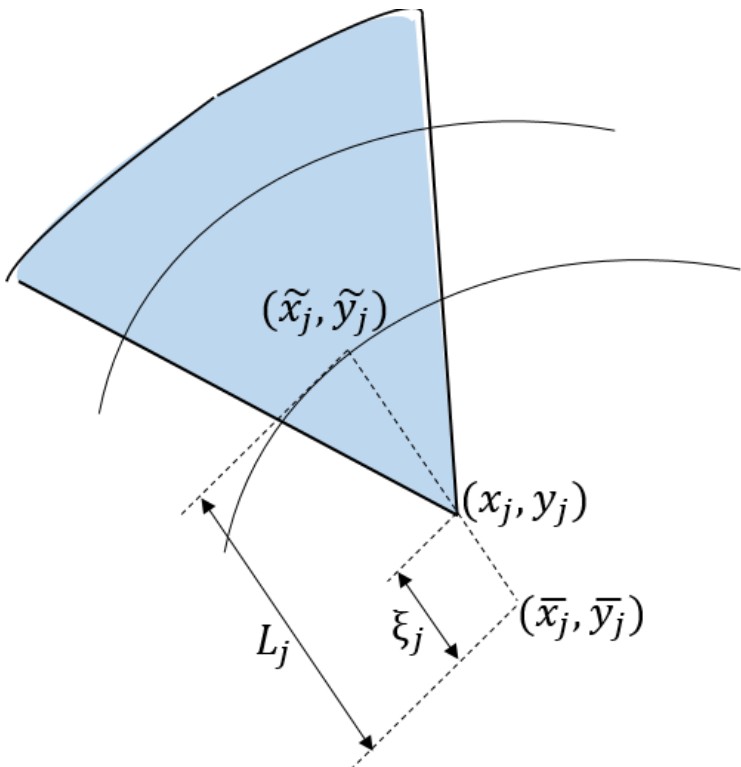

**Figure 14.** $\xi_j \in [0, L_j]$ is the varying variable for finding approximate maximum coverage in given sensing area.

## 5. Conclusions

This paper has addressed the general problem of WSN surveillance security by proposing a minimum exposure path search problem and a sensor deployment problem. Sensors, obstacles, and the target surveillance area have been modeled, and computationally efficient solutions have been proposed. For the minimum exposure path problem, the sensing limits of the sensors and the obstacle locations are used to form a "feasible area", within which the search is constrained. Vertical and horizontal grid lines are then used in this

feasible area in order to search for the minimum exposure path in a computationally efficient manner. The proposed algorithm has computational complexity $O(n^{2.5})$, a significant improvement over previous approaches, with complexity $O(mn^3)$, where $m$ is the number of obstacles and $n$ is the number of sensors.

**Supplementary Materials:** Supplementary materials can be found at https://www.mdpi.com/article/10.3390/electronics10121504/s1.

**Author Contributions:** Conceptualization, K.K.; methodology, K.K.; software, K.K.; validation, K.K.; formal analysis, K.K.; investigation, K.K.; resources, K.K. and S.L.; writing—original draft preparation, K.K. and S.L.; writing—review and editing, K.K. and S.L.; visualization, K.K.; supervision, S.L.; project administration, S.L. All authors have read and agreed to the published version of the manuscript.

**Funding:** This research received no external funding.

**Institutional Review Board Statement:** Not applicable.

**Informed Consent Statement:** Not applicable.

**Data Availability Statement:** The data presented in this study are available in supplementary materials.

**Conflicts of Interest:** The authors declare no conflict of interest.

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
