# Peer review of "Algorithms for Finding Vulnerabilities and Deploying Additional Sensors in a Region with Obstacles"

_electronics, doi:10.3390/electronics10121504_

Round 1

Reviewer 1 Report

This paper proposes algorithms to solve the minimum exposure path search problem and the sensor deployment problem applied to Wireless Sensor Networks (WSN) surveillance security. The proposed algorithm has a computational complexity lower (O(n^2.5)) than the previous ones (O(mn^3)) and it only depends on the number of sensors (n) and not on the number of obstacles (m).

My main concern about the paper is the lack of practical examples to understand the application of the algorithms proposed in real WSN scenarios. Currently, the paper covers only the mathematical development and an example to describe the system model. I recommend to include some examples of sensors deployments and to explain the application of the algorithms proposed over them and to show a comparison with previous proposed algorithms.

On the other hand, it would be interesting to explain the extension of the algorithm to a 3-D arbitrarily-shaped guarded region, because this is a more real scenario.

Finally, I recommend updating the state of the art and the bibliography, there is no reference from the last five years.

Reviewer 2 Report

This paper has addressed the general problem of WSN surveillance security by proposing a minimum exposure path search problem and a sensor deployment problem. The paper is written very well and only minor corrections are suggested (see below). However, a final evaluation of the proposed approach compared to a grid-based search method is missing and has to be added.

Details:
- line 224: Can you give an example for an impassable obstacles.
- line 227: Can you visualize the example: fw(x,y)
- line 230: Formula: directional vector ds is in the formula d.
- line 232: Fig.2: in the figure the d must be ds
- line 234: Definition 1. Could you add the term you defined?
- line 242: sentence: "We assume that sensors cannot see the target when it is obscured by obstacles."
Can you add one or two sentences describing, what happens, when is not completely blocked (e.g. half transparent obstacles)
- line 242: use latex "Definition" for area.
- line 264: Fig.4: it is hard to see where the labels p1,p3 are attached to which line. Maybe you could use colours to make it more clear.
- line 272: [2–8] tried ... -> please add: In the paper [2-8]...
- line 306: time. [] box has to be deleted
- Fig.7: e10,e11 is crossed out. Please use colours to make it more readable
- Fig.8: the same as Fig.7
- line 347: grid. [] box has to be deleted
- line 262: complexity.  [] box has to be deleted

Round 2

Reviewer 1 Report

The authors have addressed my main concerns. The revised version can be accepted for publication.

Reviewer 2 Report

The requested changes have been integrated and therefore I agree the publication in the present form.